# Rare European Beetle *Treptoplatypus oxyurus* (Coleoptera: Platypodidae) in Managed Uneven-Aged Forests of Croatia

Ivan Žarković [1], Andreja Đuka [1], Milivoj Franjević [2,*], Kristijan Tomljanović [2], Ivica Papa [1], Tomislav Krcivoj [3] and Boris Hrašovec [2]

1 Department of Forest Engineering, Faculty of Forestry and Wood Technology University of Zagreb, Svetošimunska 23, 10000 Zagreb, Croatia; izarkovi@sumfak.hr (I.Ž.); aduka@sumfak.hr (A.Đ.); ipapa@sumfak.hr (I.P.)

2 Department of Forest Protection and Wildlife Management, Faculty of Forestry and Wood Technology University of Zagreb, Svetošimunska 23, 10000 Zagreb, Croatia; ktomljanovic@sumfak.hr (K.T.); hrasovec@sumfak.hr (B.H.)

3 Department of Forest Protection and Wildlife Management, Croatian Forest Institute, Cvjetno naselje 41, 10450 Jastrebarsko, Croatia; tomislavkrcivoj@gmail.com

* Correspondence: milivoj.franjevic@sumfak.hr; Tel.: +385-98-935-2434

**Abstract:** Bark beetle outbreak sites were analysed before sanitary logging in Gorski Kotar County during spring, summer and autumn 2021. Downed European silver fir trees were inspected for red-listed saproxylic entomofauna. Among other species, the fir pinhole borer (*Treptoplatypus oxyurus*, Dufour, 1843) (Coleoptera: Platypodidae) was observed and studied on-site and in the laboratory. Symptoms of *T. oxyurus* presence were recognised as white filamentous bites of sawdust on the bark of the fir trees and the surrounding soil. Every tree infested infested with *T. oxyurus* was measured (diameter at breast height, height/length), and its position was recorded. Segments were collected for laboratory analysis to evaluate the layout and position of *T. oxyurus* gallery system. The results showed that individual corridors of *T. oxyurus*, as a rule, never intersect, cross or connect. Each family of beetles (male, female and their offspring) lives separately in its corridor system. There were examples of corridors that were very close to each other but did not touch. *T. oxyurus* is still completely unknown to forest operatives in Croatia, who do not recognise symptoms of its occurrence.

**Keywords:** Platypodidae; saproxylic beetle; biodiversity; managed forests; European silver fir

## 1. Introduction

*Abies alba* (Mill., 1768), i.e., European silver fir (Pinales: Pinaceae), is ecologically, economically and traditionally the most important Croatian conifer species, with approximately a 35% share in the total wood stock of conifers [1]. During its management, diversity of interactions between all environmental factors is taken into account, and timber harvesting operations are carried out uniformly in all stands; some other factor (climatic, biotic)—up to this point, insignificant—may become significant, i.e., decisive for development, growth, increment and survival of silver fir trees [2]. In the 21st century, the management of uneven-aged stands of silver fir is becoming more demanding in conditions of extreme-weather (droughts, windstorms and freezing rain) and bark beetles outbreaks. According to its data, Croatian Forests Ltd., a state-owned company, in cooperation with the Faculty of Forestry and Wood Technology, University of Zagreb and the Croatian Forestry Institute, assessed that, following freezing rain in Gorski Kotar County in February 2014, an icebreaker caused damage to an area of 38,341 ha. The gross volume of wood that was damaged during the storm was 1,640,771 m$^3$ of mostly broadleaved species. Uneven-aged stands of silver fir and European spruce *Picea abies* (L./Karst.), covering an area of 8225 ha, and wood stock of 93,204 m$^3$ and 14,525 m$^3$ of silver fir and European spruce, respectively, were severely damaged. Furthermore, in December 2017, a windstorm

in Gorski Kotar County caused damage to approximately 500,000 m$^3$ of wood stock [3], and even more in silver fir and European spruce stands. Because of the large amount of damaged and downed trees and rugged terrain conditions (the pre-mountainous part with terrain slope above 33%), and despite higher road density of 21.5 km/1000 ha (compared to the Croatian average [4] of 15.43 km/1000 ha forest roads), not all damaged timber was removed from the forest stands. Research by Boerner et al. [5] suggested that there is a proportional relationship between diameter at breast height (DBH) and degree of damage after demanding weather conditions. Younger and thinner trees and branches bend, as opposed to older and thicker ones that break due to increasing stiffness. This was confirmed by Oliver et al. [6], who found that a tree of 15 m in height, with a crown diameter of 6 m, can hold up to 4.5 tonnes of ice. After considerable damage caused by extreme weather conditions, coarse wood debris in windthrow gaps can become a niche for saproxylic insect diversity. The quality of wood adequate for saproxylic beetles varies greatly according to tree species [7], exposure, decaying stage, stem diameter, bark thickness, position (laying or standing), level of moisture, and the presence of associated micro-habitats. Gaps provide saproxylic diversity opportunities in managed forests that are mostly poor in coarse wood debris [8]. Coarse wood debris is one of the major food resources of red-listed species in many countries [9]. Some specialised saproxylic species only live in large dead trunks [9], which is the case with the fir pinhole borer *Treptoplatypus oxyurus* (Dufour, 1843) (Platypo-didae), and may be concentrated in windthrow gaps, usually lacking in managed forests. It can be concluded that the number of large dead trees is crucial, rather than the total volume of deadwood in the stand. Standing dead trees are more common in gaps than in intact managed stands. Snags with a higher share of red-listed species than in lying stems were found in Norwegian spruce forests [10] and Swedish deciduous forests [11]. *T. oxyurus* is a recently discovered species in Croatia, so far relatively unexplored and with little data from the literature, mainly due to its rare occurrence. *T. oxyurus* is a beetle (Coleoptera) from the subfamily Platypodinae, found within the family Curculionidae [12–14]. Sometimes *T. oxyurus* is placed as an independent family Platypodidae [15–17]. *T. oxyurus* is 4.5–5 mm long, about 1 mm narrow, and cylindrical with an elongated body. It is brown to dark brown. Sexual dimorphism on the elytrae is well-defined. The male at the hindquarters has split ends, stretched into spikes, serrated at the inner edge. The female has evenly rounded horns at the end of the elytra [18]. The head is broadly flattened at the front, with laterally placed eyes and thickened club-shaped antennae. On the dorsal part of the pronotum, there are special organs-mycangia. They serve to transmit the hyphae and the spores of fungi. In females, they are large and well-developed, and in males, they are smaller and stunted, so it is most likely that the female is mostly responsible for the transmission of fungi. These structures contain secretory glands that maintain fungal spores under favourable conditions during flight and adult movement [19]. *T. oxyurus* is present in the Pyrenees, Spain, Corsica and Sardinia, southern Italy, Greece (possibly Turkey and Iran) and India [18,20–23]. According to Balachowsky [21], in the Nearctic and Palearctic regions we encounter only a few species belonging to the genus *Platypus* (and *Treptoplatypus*); their range is often intermittent, confirming their relictual distribution. *T. oxyurus* is scarce in Central Europe, and is found in Slovakia and Germany [18]. It is intriguing and interesting that it has not yet been found in Croatias' neighbouring countries (Slovenia, Bosnia and Herzegovina), nor in the wider environment (Austria, Switzerland, the Czech Republic, Poland, Romania, Bulgaria, Albania and Macedonia), where silver fir naturally grows, and with an active entomology community well-acquainted with this group of insects [24]. Its biology is related exclusively to silver fir trees (Pinales: Pinaceae) and Greek fir *Abies cephalonica* (Loudon). Until recently, it was believed that *T. oxyurus* occurs only in the fir stands of the preserved rainforest structure. The first finding of *T. oxyurus* in Croatia was in 2010 in the area of the Northern Velebit National Park. One male *T. oxyurus* was found in one of the pheromone traps in the monitoring system of spruce bark beetle populations (European spruce bark beetle *Ips typographus* (L. 1758) (Coleoptera: Curculionidae) and six-toothed spruce bark beetle *Pityogenes chalcographus* (L.) (Coleoptera: Curculionidae) [24].

Furthermore, two fir logs densely populated with larvae and mature beetles were discovered in the Krasno area (also located in the Northern Velebit National Park) in the autumn of 2011 [25]. Today, we know that *T. oxyurus* is present in the National Parks of Risnjak, Northern Velebit and Plitvice Lakes. This paper aims to present new insights related to the development cycle, gallery systems and the structure of selected managed uneven-aged stands where *T. oxyurus* was found in 2021 following climatic extremes, which led to the spreading of this rare European saproxylic entomofauna representative.

## 2. Materials and Methods

The presence of saproxylic insect diversity was investigated in windthrow gaps and bark beetle outbreak sites of managed uneven-aged stands of silver fir and European spruce in the Delnice Forest Administration. Downed and standing trees were checked for the presence of *T. oxyurus* after the first findings in that area, and data gathering took place from June to October 2021. A Garmin 66s GPS device was used to mark the locations of *T. oxyurus*-infested trees. A Haglöf caliper was used for measuring the DBH of infested trees. The Haglöf Vertex 5 hypsometer was used to measure the height of standing infested trees, and a measuring tape was used for measuring the length of downed (windthrown) trees. During determination of stand damage after climatic extremes, it is important to define the volume of damaged trees, especially of those which will not recover, but will have to be removed from the stand [26]. The level of tree damage at which their economically justified recovery is possible is difficult to define, and depends on tree species, age of trees, management, and previous health condition etc. [27].

Samples from infected trees were collected for laboratory analysis, and for studying gallery system morphology. Samples were shaped using chisels and axes, making them suitable for ocular observation of gallery systems. Samples were made as ring cuts from infested fir logs ten centimetres thick. These cylindrical rolls were then cut into smaller pieces of circular clips. Furthermore, these clips were split into smaller segments for more straightforward observation of the layout and position of *T. oxyurus* gallery systems. Using a nylon thread, which could run through the gallery system (Figure 1), the entrance and exit holes were established, and the whole gallery system was subsequently drawn. Detailed observation of development stages and their period of activity in the gallery systems was conducted on five samples, of approximately 20 dm$^3$ volume, at room temperature (23 °C) until the development of the adults.

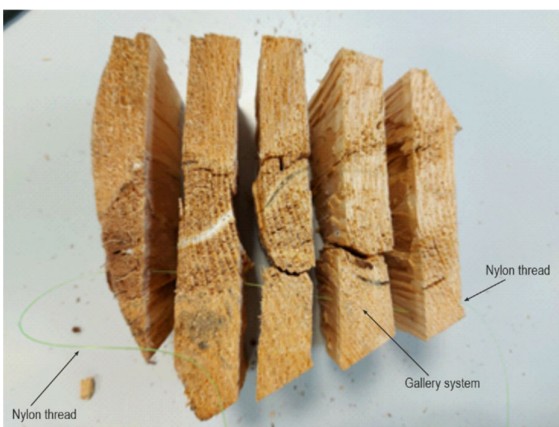

**Figure 1.** Nylon thread pulled through *T. oxyurus* gallery system.

## 3. Results

*T. oxyurus* was found in five sub-compartments (21b, 26a, 26b, 36b and 38b) in the total area of 108.95 ha in Dinaric beech and silver fir forests *Omphalodo-Fagetum* Marinček, et al., 1992 (Figure 2A) located in the Crni Lug Forest office (φ 45.41940833 N and λ 14.74161389 E) and Delnice Forest office (φ 45.40847222 N and λ 14.76649444 E), and fifty beetles were collected. Collected adults were prepared for the entomological collection, and the larvae



were stored in Eppendorf Tubes filled with 96% ETOH for future DNA barcoding. In these five sub-compartments, during 10 years of the valid Management Plan, 8281 m$^3$ of silver fir trees were cut due to windfall, ice breakage or dieback, which is 75.90% of the total prescribed allowable cut that also includes other tree species (i.e., European beech *Fagus sylvatica* L., European spruce, sycamore *Acer pseudoplatanus* L., etc.). The downfall of silver fir trees due to windstorms accounted for 4808 m$^3$, a substantial amount in the whole share of removed trees. In July and August, fresh sawdust can be visible and indicates beetle activity in silver fir trees. Thirty-three silver fir trees infested with *T. oxyurus* were recorded, and amounted to a total volume of 171.644 m$^3$ (Figure 2B), with an average DBH at 65.27 cm.

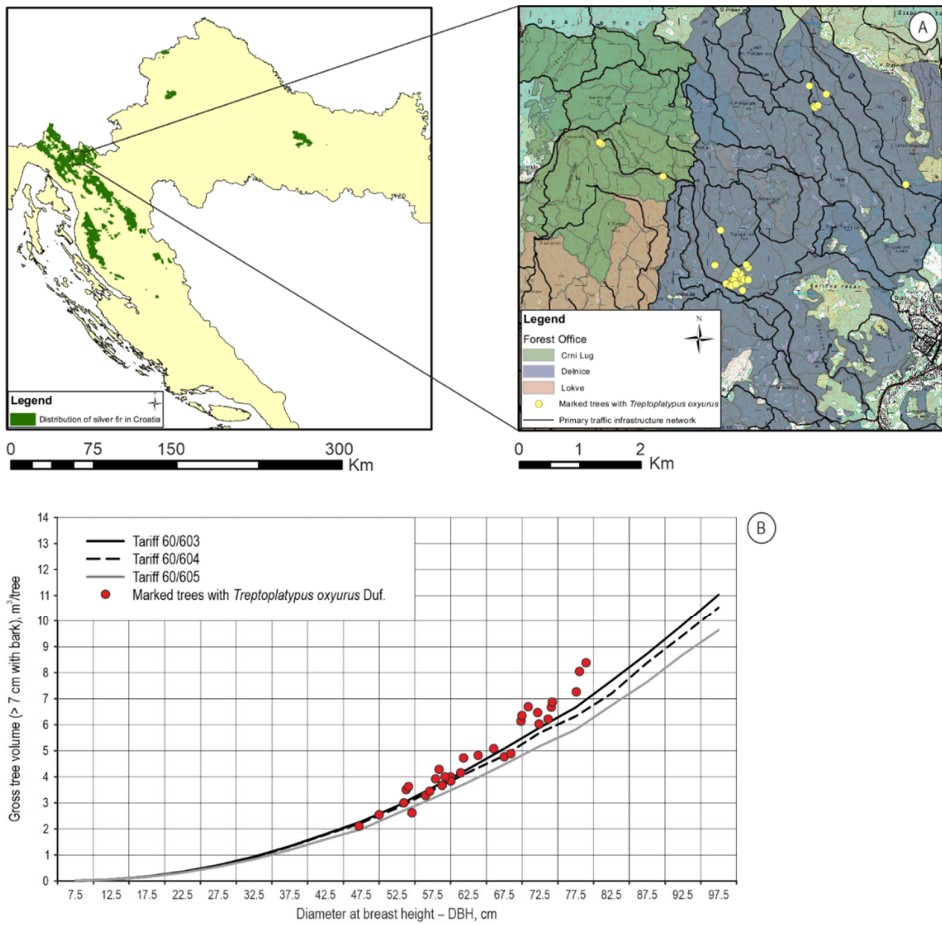

**Figure 2.** (**A**) Study site with locations of silver fir trees infested with *T. oxyurus*; (**B**) Volume of infested trees compared to the three prescribed tariffs from valid Management Plans of the area.

Figure 2B shows the range of tree sizes that the beetles infested, and is of somewhat larger volume, i.e., more than half of the infested timber volume was in the highest DBH class, which is an old age for silver fir, and a sign that these forests are old and physiologically weak.

The *T. oxyurus* gallery system was placed inside a silver fir trunk on one plane, perpendicular to the longitudinal axis of the trunk. If the tree was in a standing position, the gallery system was on the horizontal plane, and if the tree was downed/felled, the gallery system was on the vertical plane. After consecutive probing with nylon threads (Figure 1), the development of the gallery system was drawn during the period of the laboratory analysis, and the following patterns of early-, mid- and late-development stages were observed (Figure 3).

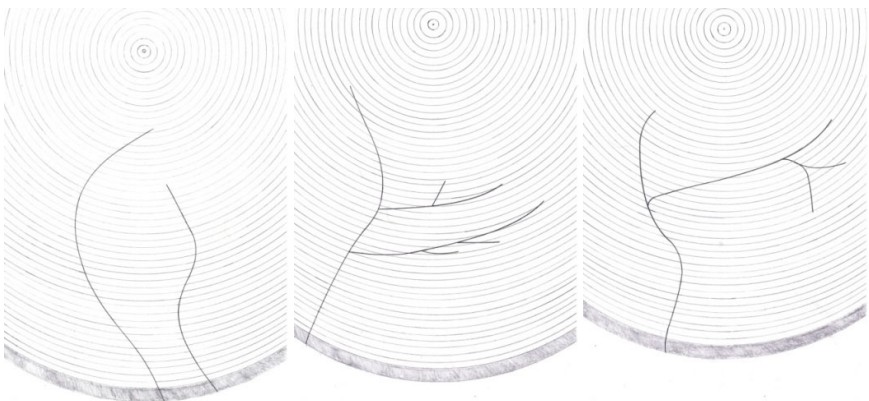

**Figure 3.** *T. oxyurus* gallery system: early-, mid- and late-development stages.

*T. oxyurus* did not strictly follow this rule when biting the gallery system; then, the beetle could ascend or descend with slight inclinations. Gallery corridors were slightly winding and mainly were directed towards the centre of the trunk. Some galleries consisted of only one main corridor, but there were also some made up of multiple corridors. In the latter case, the side corridors branched off from the central corridor, directed towards the centre of the trunk. The range of angles at which the corridors branched extended from very small pointed to almost vertical. These lateral branches of the gallery system might continue to advance toward the centre of the trunk. Furthermore, they could approximately follow the line of the growth rings and slightly intersect it. The third option was to turn back towards the bark of the tree. In addition, side branches of the corridor could be separated again (Figure 3). The corridor systems of this beetle always branched out in the direction from the entrance to the interior. In the opposite direction, towards the surface of the trunk, the corridors did not fork. Galleries had only one entrance; no two or more entrances would merge into one gallery.

Each family of *T. oxyurus* (male, female and their offspring) lived separately in its corridor system. There were examples of corridors that were very close to each other but did not touch, which means that when biting the corridor, *T. oxyurus* felt the proximity of the adjacent corridor and moved to the other side. In addition to horizontal corridors, there were also vertical corridors. Vertical corridors were parallel to the axis of the trunk, and could be directed towards the top or towards the tree's core. Vertical corridors were short, about 1 cm long, and were found in lateral horizontal corridors parallel to the growth lines. Vertical corridors could be located both in the middle and at the end, but were rare and difficult to find (Figure 4). Short vertical corridors were bitten by larvae in the last developmental stage, before pupation, and this was the location where they then pupated. During the pupal stage, the corridor was closed, and there was a partition made of sawdust at the entrance. Also, a slight curvature at the junction with the other corridors enabled *T. oxyurus* easier exit in the adult stage. The corridors of the *T. oxyurus* mainly were black. This was from the fungi that covered the sides of the corridor. As *T. oxyurus* is a saproxylic and xylomicetophagous insect, adult individuals bring fungal spores on their bodies that further multiply and gradually spread through the gallery system [18].

There was a dark layer of fungi in previously bitten corridors or parts of corridors. As the mycelium grew, it often spread to the surrounding wood, becoming dark in colour. Few dark walls were found in the early stages of corridor development, where fungi had not yet developed. The fungus was also absent in short vertical corridors. In old, unmaintained and abandoned corridors, the fungus multiplied and grew so much that it eventually closed the corridors with its mycelium. The fungi myceliums were not black but white, and wadded in texture (Figure 5).

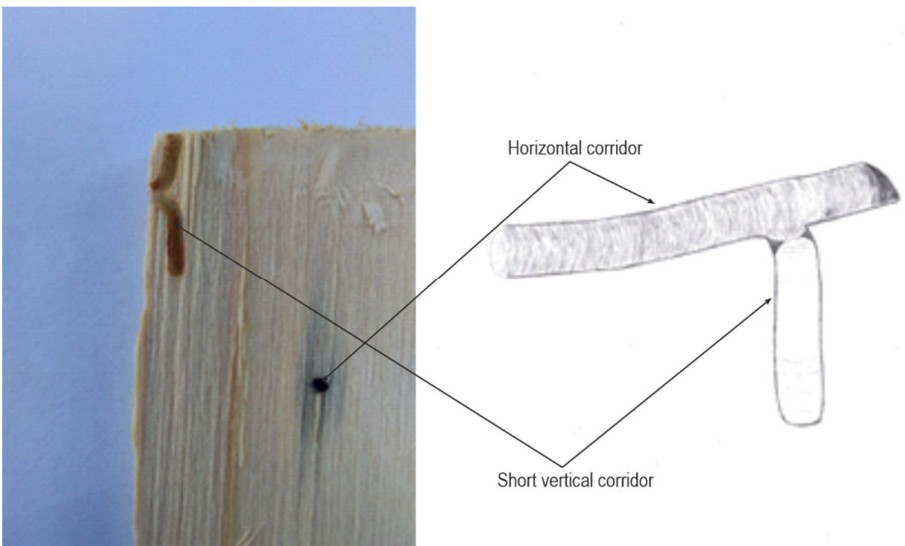

**Figure 4.** Short vertical corridors directed upwards and downwards at the end of the horizontal corridor.

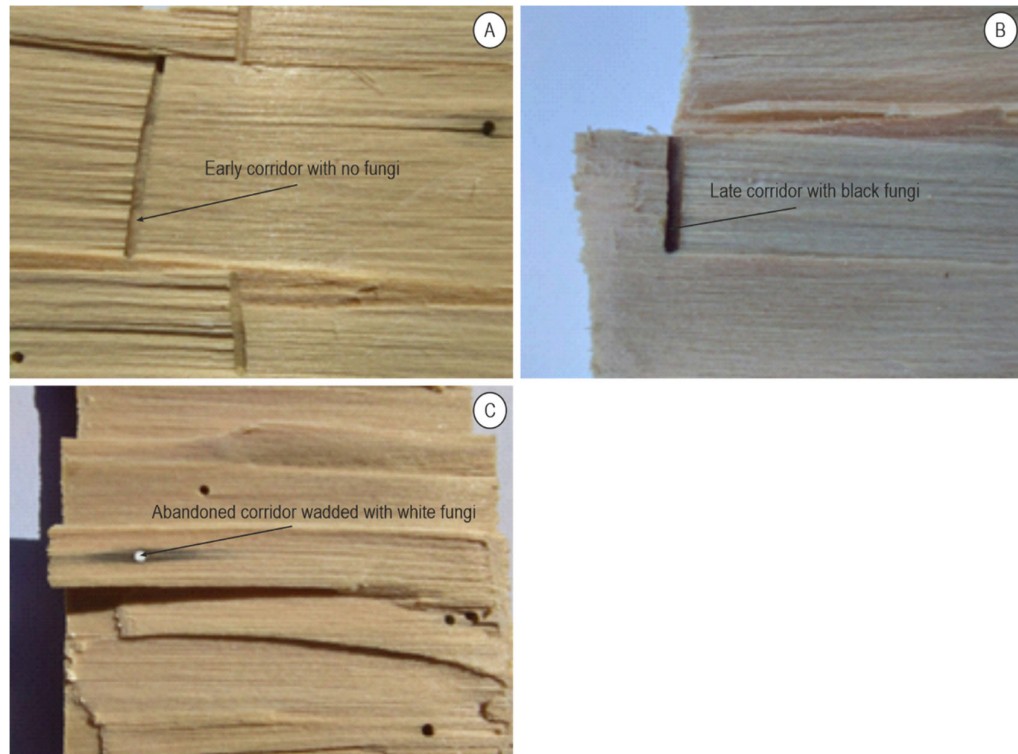

**Figure 5.** Stages of corridor development and fungus development: (**A**) early; (**B**) late; and (**C**) abandoned corridor.

The life of the *T. oxyurus* took place mostly in the gallery systems within the fir trunk. During a swarming period, part of their lives was spent outside of their secret shelter. Multiple observations in the uneven-aged managed stands showed that the swarming period was in the summer months, usually during July and August. *T. oxyurus* is a monogamous species. After successful copulation, *T. oxyurus* started to burrow into the fir trunk. Like all saproxylic beetles, they attack already physiologically weakened trees, dead and decaying trees, trees under stress and freshly fallen or felled trees. Symptoms of *T. oxyurus* infestation were recognised on the bark of fir trees and the surrounding soil as white filiform sawdust, unique in shape and size for this genus, differentiating it from

species such as ambrosia beetles, bark beetles, longhorn beetles and weevils. Symptoms were noticeable during August, September and October, during drilling of the beetle into the trunk and biting of the gallery system (Figure 6).

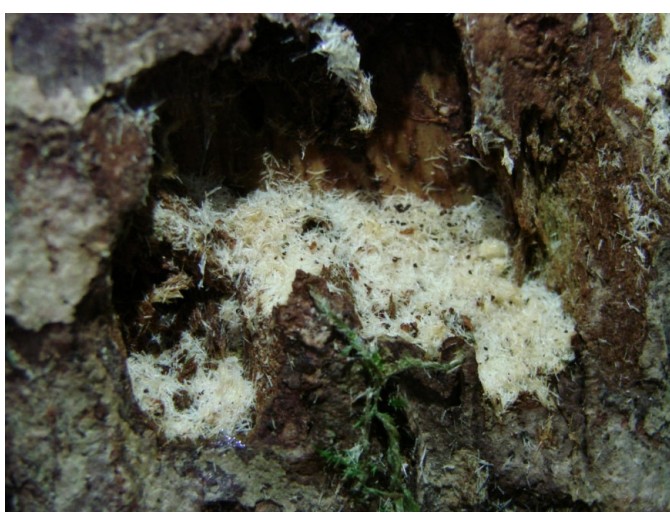

**Figure 6.** Filiform sawdust on the bark of silver fir tree–the symptom of *T. oxyurus* infestation.

Drilling and biting the corridors took place, like the oak pinhole borer *Platypus cylindrus* (Fabricius, 1792), where the female leads the drilling. With her strong mandibles, the female bit off pieces of wood, and slowly entered the interior of the trunk. The female was followed by a male who threw bites of sawdust out of the trunk. Each gallery system had a one-entrance opening. Entry of *T. oxyurus* into the silver fir wood consisted of the main parent corridor, which extended from the entrance on the bark towards the centre of the trunk. With the entry of these insects into the fir wood, infection of the wood itself with fungal spores occurred. During the brooding, the female laid eggs. After some time, pale yellow, shiny larvae emerged from the eggs, which, like the adults, fed on the mycelium of the fungus. The larvae also had well-developed mandibles on their heads, with which they gnawed their corridors. By the following spring, parental adults, eggs and larvae could be found in the gallery systems (Figure 7), which could mean that the female laid eggs until that spring. From spring to June, parental adults and larvae of all developmental stages could be found in the galleries (Figure 8). After the larvae had sufficiently developed and reached the last larval stage, in the lateral corridors that approximately followed the growth ring line, they began to gnaw short corridors perpendicular to the previous ones and parallel to the longitudinal axis of the trunk. Before pupation, they turned their heads towards the exit, which they closed with grits, and thus made a pupating cradle. *T. oxyurus* has a pupa libera. Pupation began in early July, and in mid-July the first adults of the new generation appeared. At that point, all developmental stages were present except the eggs: parental adults were dark brown, larvae before pupation, pupae and young adults were light brown. A new generation emerged from the fir wood in search of a mate and a suitable host to produce their offspring (Table 1). During the observation of the *T. oxyurus* life cycle, it was observed that adult individuals of this species possess the property of an elytro-tergal type of stridulation. Stridulation is the ability to produce sound or vote by rubbing one part of the body against another. This property helps individuals to find their partners, communicate with each other, and produce sound if they are in danger.

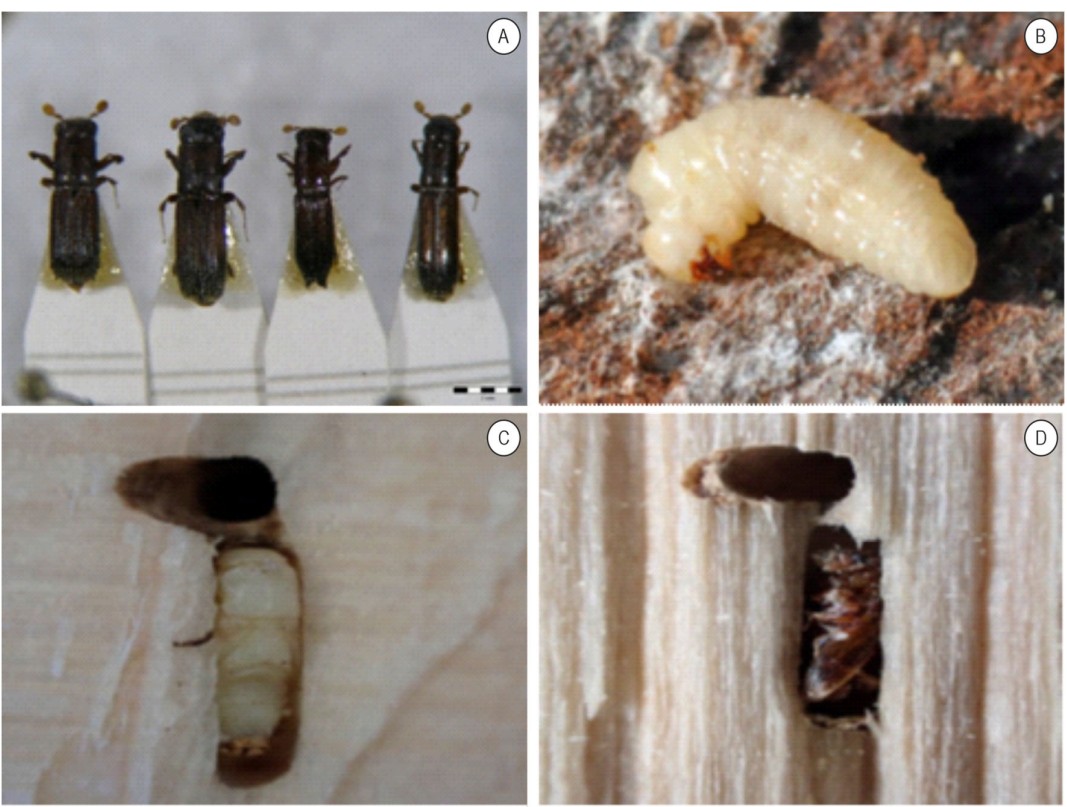

**Figure 7.** (**A**) Adults of *P. cylindrus* and *T. oxyurus* (from left to right *P. cylindrus* ♂, *P. cylindrus* ♀, *T. oxyurus* ♂, *T. oxyurus* ♀); (**B**) larva of *T. oxyurus*; (**C**) pupa libera of *T. oxyurus*; (**D**) young adult of *T. oxyurus* inside gallery system.

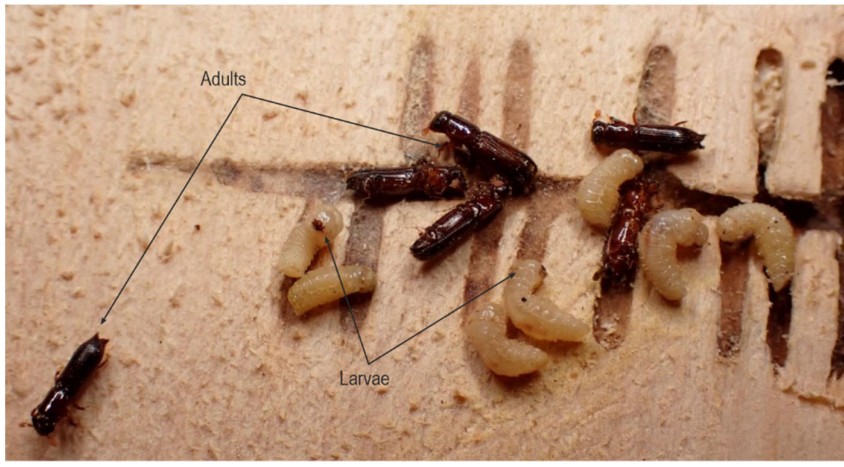

**Figure 8.** Gallery system with larvae and young adults of *T. oxyurus*.

**Table 1.** Life cycle and development stages of *T. oxyurus* (Legend: ● egg; 0 larva; ◇ pupa; + adult).

| YEAR/MONTH | JAN | FEB | MAR | APR | MAY | JUN | JUL | AUG | SEP | OCT | NOV | DEC |
|---|---|---|---|---|---|---|---|---|---|---|---|---|
| 1 |  |  |  |  |  |  | ●+ | ●0+ | ●0+ | ●0+ | ●0+ | ●0+ |
| 2 | ●0+ | ●0+ | ●0+ | ●0+ | 0+ | 0+ | ◇0+ | ◇+ |  |  |  |  |

## 4. Discussion

In the Croatiaon forests of silver fir—the most essential coniferous tree species of Croatian forestry, and the most endangered tree species—climatic extremes followed by beetle outbreaks resulted in the recent more-substantial decline and dieback of silver fir.

*T. oxyurus* was recently discovered in Croatia, but only in the protected forests of the Northern Velebit and Risnjak National Parks. *T. oxyurus* is still completely unknown to forest operatives in Croatia, who do not recognise symptoms of its occurrence (Figure 6) even after windthrow gaps or *I. typographus* outbreaks, nor during sanitation felling.

Two species of the subfamily Platypodinae inhabit Europe. The oak pinhole borer *Platypus cylindrus* (Coleoptera: Platypodidae) and *T. oxyurus* (Figure 7). *P. cylindrus* is considered a pest in Croatian forestry, and a species which damages the most valuable veneer assortments of oak roundwood. This pinhole borer never occurs on a large scale, but only in small places, where the damage is such that it completely destroys the commercial value of the assortment [23]. In the 20th century, the perception of *T. oxyurus* was the same as the perception of *P. cylindrus* today in Croatia, i.e., it was considered a pest. It has been suggested that if the population of *T. oxyurus* continues to increase, certain protection measures will need to be taken [18]. The author goes on to say that it is necessary to cut trees that are susceptible hosts, thus enabling further population growth. The same author states that, given the swarming period in July and August, dead firs should be cut down and removed no later than mid-June, especially from the edges of stands. Logs that may have remained in the forest stands or at the roadside landing sites, after felling performed in autumn and winter, should also be removed.

Changes in abundance and distribution patterns of resources on macro- and micro-habitat scales affect the abundance and distribution of insect species. Intact and salvage-logged gaps also differ. Forest managers often harvest timber for timber trade and forest protection in wind gaps. The ecological changes reported above are different in salvaged gaps, and the environmental heterogeneity is reduced. The removal of all fallen trees reduces the coarse wood debris volume and diversity. Harvesting can result in soil scarification, damage of uprooted trees and reduction of the micro-topography, and the development of a dense cover of grass which in turn suppresses most bryophytic species [28]. In managed forests, residual standing trees are also regularly removed, leading to a difference in mean light level between uncleared (29%) and cleared (50%) gaps [29]. In salvaged gaps, the wood that has not been removed is often fragmented and entirely on the ground, where it decays more rapidly. Weslien and Schröter reported an outbreak that killed 16% of spruce trees during the seven years following windthrow in a stand of 105 ha [30]. This outbreak collapsed without any human counteractive measures due to natural enemies of *I. typographus*. After windthrow gaps and bark beetle outbreaks, *T. oxyurus* is an abundant species and not a pest in the managed uneven-aged stands of Croatia, but a valuable element of biodiversity and an interesting representative of saproxylic entomofauna. Many aspects of its biology and ethology are still unexplored, such as an elytro-tergal type of stridulation observed during this research. Although the morphological characteristics of stridulation organs are well-represented in the scientific literature, very little is known about acoustic behaviour and acoustic reception in Coleoptera species [31,32]. One example within platypodines revealed the importance of close-range sound communications of the oak platypodine beetle *Platypus quercivorus* (Murayama 1925) (Coleoptera: Curculionidae), in which females emit sounds to invite males to come, and upon arrival in the new gallery, males emit sound signals and thus "mark" their territory [33]. Within the corridor systems of *T. oxyurus*, complex social behaviour and structure are observed (Figure 8) during interaction at multiple development stages and multiple age classes of siblings in a single gallery system. It is interesting and important to say that the individual corridors of *T. oxyurus*, as a rule, never intersect, cross or merge.

During the developmental cycle, the male helps the female clean the sawdust from the hallway and protects the entry of predators and parasitoides [34]. Insects that adapted and developed specialised organs-mycangia provided the fungus-protection from external conditions, and enabled transport to the host. It is assumed that climatic and habitat conditions affect its distribution, and the existence and availability of symbiotic fungi. The diversity of fungicolous wood-living beetles (Ciidae, Cryptophagidae, Latridiidae, Erotylidae, Mycetophagidae) and flies (Mycetophilidae) seems to be at its greatest about one

decade after the death of the tree [35]. Windthrow gaps induce a multi-sized distribution of open patches, unlike artificial openings. They also provide deadwood in different stages of decay and exposure levels, and ensure "log continuity" [36] as well as a more general temporal and spatial continuity of suitable habitats [37]. During the first five years after a storm, species abundance and richness were slightly higher in unsalvaged than in salvaged gaps where harvested timber served as a deadly trap for saproxylic, predators and parasitoids [38].

Due to the occurrence of *T. oxyurus* in the managed forest stands of Croatia, signs of stand preservation and biodiversity are evident. *T. oxyurus* is an important biotic factor in the natural regeneration process of old and weakened fir trees, especially after weather calamities [26] and bark beetle outbreaks, which have been common in the last decade in the forests of Gorski Kotar. This research showed that windfall disturbance can be an important agent that creates habitat heterogeneity in time and space, thus being a driving force of forest succession and a source of regional biodiversity in forest ecosystems.

**Author Contributions:** Conceptualisation, B.H., M.F., A.Đ. and I.Ž.; methodology, I.Ž., A.Đ. and M.F.; software, K.T.; validation, A.Đ., I.P. and K.T.; formal analysis, T.K., A.Đ. and M.F.; investigation, I.Ž., A.Đ., I.P. and M.F.; writing—original draft preparation, M.F.; writing—review and editing, T.K. and A.Đ.; visualisation, M.F. and B.H.; supervision, A.Đ.; project administration, A.Đ.; funding acquisition, A.Đ. All authors have read and agreed to the published version of the manuscript.

**Funding:** This research was funded by the Croatian Science Foundation under the project "Quantity and structure of fir and spruce biomass in changed climatic conditions" (UIP-2019-04-7766).

**Informed Consent Statement:** Not applicable.

**Data Availability Statement:** Not applicable.

**Acknowledgments:** The authors thank the Forest Administration Office Delnice (Croatian Forests Ltd.) for support during field activities and data gathering.

**Conflicts of Interest:** The authors declare no conflict of interest. The funders had no role in the design of the study; in the collection, analyses or interpretation of data; in the writing of the manuscript; or in the decision to publish the results.

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
