# Peer review of "Rare European Beetle Treptoplatypus oxyurus (Coleoptera: Platypodidae) in Managed Uneven-Aged Forests of Croatia"

_forests, doi:10.3390/f13040580_

Round 1

Reviewer 1 Report

Dear authors,

I am grateful to you for taking into account all the recommendations in the new version of the manuscript.

Author Response

Reviewer 1

Thank you for all your comments.

Reviewer 2 Report

This manuscript documents the natural history of a previously understudied xylomycetophagous beetle, Treptoplatypus oxyurus, in Croatia. Comparative entomologists, forest managers, and community ecologists will all find the data and information presented in this paper useful. 

Overall, I found the most useful part of the paper to be the documentation of the natural history of the beetles. The description of the morphology, the gallery formation, and the type, style, and age of wood being infested were all valuable information. 

Weaknesses of this manuscript are primarily the links between the different sections. For example, the underdevelopment (in the introduction, methods, and results) of the importance of natural windfalls in this situation do not support the extensive discussion on this topic. 

The discussion in particular seems a jumble of ideas that do not necessarily follow. I would urge the authors to re-write this section with more clarity and more intentionality. A large amount of the general discussion is not even derived from the results of the paper and seems to wander through half-formed ideas about biodiversity, forest management, and successional aspects of windfall gaps. If this is indeed to be the point of the paper, then this should be made clear in the introduction and the methods/results should reflect this. As currently written, the methods and results are much more indicative of a detailed natural history exploration of an interesting and previously unrecorded xylomycetophagous beetle in Croatia. The discussion should more generally reflect this. The discussion as it stands could be a final paragraph discussing why such natural history explorations are necessary: they provide information to forest managers regarding the signs of infestation; they are important data for understanding the consequences of climate-induced shifts in forest community structure; and they set the stage for more detailed investigations in the roles of forest complexity (and therefore forest management) into the maintenance of biodiversity.

Overall the English language and style are good, although at times I found myself struggling to understand what the object of the sentence was without careful re-reading. I have provided some suggestions that I think will decrease this issue and they are noted below along with editorial comments.

I think the figures are all useful but could be annotated for more utility: Figure 1 could use an arrow to indicate the nylon thread and the gallery; Figure 2a could include an inset to indicate where in Croatia this quadratic is located (and perhaps even a distribution of silver fir in Croatia); Figure 2b needs clarification indicated in my comments below; in Figure 3 it is difficult to discern how early, mid, and late development stages correspond to anything in the text as it appears that these represent initial unbranched and then two variations on the types of branching described; Figures 4 & 5 could use annotations (e.g. arrows or circles) to direct the reader's attention to the relevant structures; Figure 6 is good; Figure 7 is good (although images that allow examination of relevant characters instead of overall habitus would be more useful for discriminating the two species); Figure 8 could use some expansion on why what we are looking at is relevant either in the figure legend or in the text (I discuss this in the comments below).

Line 38: “stands of silver fir”

Line 39: “droughts”, not “draughts”

Lines 41-44 read awkwardly. Was the damage 38,341 ha and impacting 1,640,771 m3 of wood stock? Or was this the area within which considerable damage occurred. This same line of difficulty reading continues through line 46.

Line 53: “after calamities and extreme weather conditions” reads awkwardly. Perhaps “after considerable damage caused by extreme weather conditions”?

Lines 70-71 sentence “Some authors believe…” can be removed.

Lines 71-73 are confused. The first clause refers to the subfamily (inhabit tropical and subtropical areas), the second to T. oxyurus (Mediterranean area). This should be clarified.

Line 79: “mikangii” can be translated to “mycangia”.

Line 87: change “relict character” to “relictual distribution”

Line 112: change “June to and October 2021” to “June to October 2021”. Begin the next line with “A Garmin 66s GPS…”

Line 113: Add indefinite article “A Haglöf caliper…”

Line 114: Change to indefinite article “A Haglöf Vertex…”

Figure 2a: Perhaps an inset as to where this quadrat is within Croatia?

Figure 2b: Perhaps the “prescribed tariffs” should be explained. Presumably these are the trees that were being allowed for removal. Also, what is the table at the base of the figure. It is not explained and has no legend. Do these refer to the actual points of the three different tariffs to create the slopes? Why is this necessary. Also, what, overall, is the point of this figure. Is it to show the range of tree sizes that the beetles infest? Is it to show that the beetles tend to infest trees with somewhat larger volume for DBH than what is generally present in the forests?

Lines 149-151 do not lead necessarily to the following discussion of the gallery system.

Line 149: “It can be concluded that more than…”

Line 186: “xylomycetophagous”

Figures 4 & 5: I would urge the authors to use arrows on the figures to highlight the futures indicated in the photographs.

Line 219: Add definite article: “With her strong mandibles, the female bit off…”

Line 240: Add definite article: “During observation of the…”

Lines 241-244. Was the stridulatory structure observed? Is it mandibular? Metafemoral? Could this be imaged or figured? The article remarks later (line 286) that this is elytro-tergal. Why not include that here?

Line 253: Slovakian author? Why is the nationality of the author important? This should be about the location of the study. E.g. “In Slovakia, it is suggested that if the population of T. oxyurus is increasing, certain protection measures need to be taken [16].”

Line 279 should indicate up front that this paper is about an outbreak of Ips typographus. As written is comes off as being an outbreak of T. oxyurus.

Lines 289-298 are awkward as it is difficult to figure out which statements refer to T. oxyurus and which to P. quercivorus. This should be clarified In addition, the “For instance” modifies the previous “very little is known”. Perhaps change this to “One example within platypodines revealed the importance of close range sound communications…” or something like that. Finally, what were the complex social behavior and structure observed that we are supposed to witness in Figure 8? That multiple generations interact? That there are multiple age classes of siblings in a single gallery? PLEASE expand. 

Line 299: change to “mycangia organs” or just “mycangia”.

Author Response

Reviewer 2

Thank you for all your comments and corrections. All changes that we have done are highlighted in red. Please see the attachment.

Reviewer 2

Thank you for all your comments and corrections. All changes that we have done are highlighted in red.

Specific comments of the reviewer:

Line 38: “stands of silver fir” CORRECTED

Line 39: “droughts”, not “draughts” CORRECTED

Lines 41-44 read awkwardly. Was the damage 38,341 ha and impacting 1,640,771 m3 of wood stock? Or was this the area within which considerable damage occurred. This same line of difficulty reading continues through line 46. CORRECTED

Line 53: “after calamities and extreme weather conditions” reads awkwardly. Perhaps “after considerable damage caused by extreme weather conditions”? CORRECTED

Lines 70-71 sentence “Some authors believe…” can be removed. CORRECTED

Lines 71-73 are confused. The first clause refers to the subfamily (inhabit tropical and subtropical areas), the second to T. oxyurus (Mediterranean area). This should be clarified. CORRECTED

Line 79: “mikangii” can be translated to “mycangia”. CORRECTED

Line 87: change “relict character” to “relictual distribution” CORRECTED

Line 112: change “June to and October 2021” to “June to October 2021”. Begin the next line with “A Garmin 66s GPS…” CORRECTED

Line 113: Add indefinite article “A Haglöf caliper…” CORRECTED

Line 114: Change to indefinite article “A Haglöf Vertex…” CORRECTED

Figure 2a: Perhaps an inset as to where this quadrat is within Croatia? ADDED

Figure 2b: Perhaps the “prescribed tariffs” should be explained. Presumably these are the trees that were being allowed for removal. Also, what is the table at the base of the figure. It is not explained and has no legend. Do these refer to the actual points of the three different tariffs to create the slopes? Why is this necessary. Also, what, overall, is the point of this figure. Is it to show the range of tree sizes that the beetles infest? Is it to show that the beetles tend to infest trees with somewhat larger volume for DBH than what is generally present in the forests? ADDED AND EXPLAINED

Lines 149-151 do not lead necessarily to the following discussion of the gallery system. CORRECTED

Line 149: “It can be concluded that more than…” CORRECTED

Line 186: “xylomycetophagous” CORRECTED

Figures 4 & 5: I would urge the authors to use arrows on the figures to highlight the futures indicated in the photographs. ADDED

Line 219: Add definite article: “With her strong mandibles, the female bit off…” CORRECTED

Line 240: Add definite article: “During observation of the…” CORRECTED

Lines 241-244. Was the stridulatory structure observed? Is it mandibular? Metafemoral? Could this be imaged or figured? The article remarks later (line 286) that this is elytro-tergal. Why not include that here? ADDED

Line 253: Slovakian author? Why is the nationality of the author important? This should be about the location of the study. E.g. “In Slovakia, it is suggested that if the population of T. oxyurus is increasing, certain protection measures need to be taken [16].” CORRECTED

Line 279 should indicate up front that this paper is about an outbreak of Ips typographus. As written is comes off as being an outbreak of T. oxyurus. CORRECTED

Lines 289-298 are awkward as it is difficult to figure out which statements refer to T. oxyurus and which to P. quercivorus. This should be clarified In addition, the “For instance” modifies the previous “very little is known”. Perhaps change this to “One example within platypodines revealed the importance of close range sound communications…” or something like that. Finally, what were the complex social behavior and structure observed that we are supposed to witness in Figure 8? That multiple generations interact? That there are multiple age classes of siblings in a single gallery? PLEASE expand. EXPLAINED

Line 299: change to “mycangia organs” or just “mycangia”. CORRECTED

Reviewer 3 Report

This manuscript brings an interesting description on the natural history of Treptoplatypus oxyurus in Croatia. According to the manuscript, this ambrosia beetle is rare in Europe and no in-depth study on the biology of this insect is known. From the observational data, it is concluded T. oxyurus is not a pest in Croatia. 

Although this manuscript has its value on the original descriptions of the natural history, I feel the results could be better presented (see below). Also, the methods used to describe the natural history of the beetle should be better stated in the manuscript.

General comments:

1.  Due to the nature of this manuscript (natural history of a bettle) and the way its currently written, I think one approach that could benefit the manuscript is merging both Results and Discussion sections. This will probably avoid repetition of statements made in both sections. In addition, it will help shorten the manuscript.

2. Details on the methods used for describing the natural history of T. oxyurus are missing. For exemple, in L. 123-125), how were these observations made? In L. 133 it is stated that 50 beetles were collected. There is no info regarding to what was done with these beetles. Also, when describing the results, it would interesting to inform how many eggs per chamber were found and how many adults emerged during the observational time.  

3. Although the observation on the stridulation behavior is interesting, there is no empirical evidence to support if this behavior indeed induced gallery construction (avoidance) in T. oxyurus. This conclusion is a bit overstated in the manuscript and should be left as an speculation in the discussion only (and should not appear in the abstract, L. 26-27).

4. L. 282-286: I think this is an important message of this manuscript and also an interesting finding. This should be highlighted in the abstract and should be one of the main conclusion of this paper. 

Specific comments:

L. 14-15: No need to mention the project name in the abstract, just use a general term like "As part of a forestry survey..." or a similar wording. 

L. 21-22: Delete "with a GPS device". This is too specific to include in the abstract.

L. 23: "and subsequently a gallery system was drawn". This is also too specific to mention in the abstract.

L. 38: Replace "and" for "of".

L. 57: I think a fungus can be considered a micro-habitat in a microscopic level? Please, rephrase this sentence or delete the word "fungi".

L. 66-67: This sentence would be better rephrased as "T. oxyurus was a recently discovered species in Croatia,..."

L. 69-70: Move this sentence above when mentioning the beetle for the first time to avoid repetition.

L. 79: Replace to "mycangia"

L. 81: I'm aware it was not the aim of this study, but which fungus does this beetle possibly vector? It would be interesting to mention references for the putative fungus (name) in the manuscript.

L. 112: "To" or "and" for "June to and October"?

L. 174-175: This seems more an interpretation or speculation of the observed phenomena, so this sentence should be in the discussion section.

L. 213: delete "bark" after "ambrosia".

L. 242-243: Because this behavior is defined here, please, cite references to support the definition.

L. 253: "An Slovakian"

L. 254: "continues to increase"

L. 255: replace "fell" to "cut".

L. 259: should feelings be replace by "seasons"?

Table 1: I feel there is no need of this Table. The contents can be easily be inserted as a paragraph in the results section.

L. 297: This sentence is not clear: it is similar to what?

L. 316: remove "s" from "saproxylics".

Figure 2 could be cited in the introduction section (as Figure 1) to show the study area.

Figure 4 and 5: I suggest merging both figures, so there will be one figure related to tunneling system in this species.

Figure 7: This figure should be cited in the results section, specially  where the biology of T. oxyurus is described.

Figure 8 could easily replace Fig 7b, so that decreasing the number of figures in the paper.

Author Response

Reviewer 3

Thank you for all your comments and corrections. All changes that we have done are highlighted in red. Please see the attachment

Specific comments of the reviewer:

  1. 14-15: No need to mention the project name in the abstract, just use a general term like "As part of a forestry survey..." or a similar wording. CORRECTED
  2. 21-22: Delete "with a GPS device". This is too specific to include in the abstract. CORRECTED
  3. 23: "and subsequently a gallery system was drawn". This is also too specific to mention in the abstract. CORRECTED
  4. 38: Replace "and" for "of". CORRECTED
  5. 57: I think a fungus can be considered a micro-habitat in a microscopic level? Please, rephrase this sentence or delete the word "fungi". CORRECTED
  6. 66-67: This sentence would be better rephrased as "T. oxyurus was a recently discovered species in Croatia,..." CORRECTED
  7. 69-70: Move this sentence above when mentioning the beetle for the first time to avoid repetition. CORRECTED
  8. 79: Replace to "mycangia" CORRECTED
  9. 81: I'm aware it was not the aim of this study, but which fungus does this beetle possibly vector? It would be interesting to mention references for the putative fungus (name) in the manuscript. Because of the complexity and the number of fungi species that we have found in T. oxyurus corridors and ongoin laboratory analysis this was not the scope of this research, but will be presented in the future.
  10. 112: "To" or "and" for "June to and October"? CORRECTED
  11. 174-175: This seems more an interpretation or speculation of the observed phenomena, so this sentence should be in the discussion section. CORRECTED
  12. 213: delete "bark" after "ambrosia". CORRECTED
  13. 242-243: Because this behavior is defined here, please, cite references to support the definition. CORRECTED
  14. 253: "An Slovakian" REPHRASED
  15. 254: "continues to increase" CORRECTED
  16. 255: replace "fell" to "cut". CORRECTED
  17. 259: should feelings be replace by "seasons"? REPHRASED

Table 1: I feel there is no need of this Table. The contents can be easily be inserted as a paragraph in the results section. Description of the life cycle in the table seems easier to follow than the writing of the whole cycle in the text, specially with overlapping developing stages of the beetle.

  1. 297: This sentence is not clear: it is similar to what? CORRECTED
  2. 316: remove "s" from "saproxylics". CORRECTED

Figure 2 could be cited in the introduction section (as Figure 1) to show the study area. Figure 2 shows the study area, but also the position and size of the infested trees, so we feel it should be placed in the result section.

Figure 4 and 5: I suggest merging both figures, so there will be one figure related to tunneling system in this species. Because of the phenomenon of short vertical corridors, which are associated with beetle etology (fig. 4), we divided it from the fungi development in corridors in fig 5.

Figure 7: This figure should be cited in the results section, specially  where the biology of T. oxyurus is described. CORRECTED

Figure 8 could easily replace Fig 7b, so that decreasing the number of figures in the paper. This is one of the rare photos in the laboratory conditions that shows multiple development stages of T. oxyurus.

This manuscript is a resubmission of an earlier submission. The following is a list of the peer review reports and author responses from that submission.

Round 1

Reviewer 1 Report

an accurate reading of the text by a native English speaker is recommended

Reviewer 2 Report

Dear authors,

Thank you for the interesting and new information on the biology and distribution of the rare beetle Treptoplatypus oxyurus. My recommendations are in the attached file.
